# Is Formal Social Participation Associated with Cognitive Function in Middle-Aged and Older Adults? A Systematic Review with Meta-Analysis of Longitudinal Studies

**DOI:** 10.3390/bs14040262

**Published:** 2024-03-22

**Authors:** Cláudia Cunha, Gina Voss, Renato Andrade, Alice Delerue-Matos

**Affiliations:** 1Communication and Society Research Centre, Institute of Social Sciences, University of Minho, 4710-057 Braga, Portugal; adelerue@ics.uminho.pt; 2Faculty of Medicine, University of Porto, 4200-319 Porto, Portugal; up202301654@up.pt; 3Clínica Espregueira—FIFA Medical Centre of Excellence, 4350-415 Porto, Portugal; randrade@espregueira.com; 4Dom Henrique Research Centre, 4350-415 Porto, Portugal; 5Porto Biomechanics Laboratory (LABIOMEP), Faculty of Sports, University of Porto, 4200-450 Porto, Portugal; 6Department of Sociology, Institute of Social Sciences, University of Minho, 4710-057 Braga, Portugal

**Keywords:** formal social participation, cognitive function, middle-aged, older adults

## Abstract

This systematic review with meta-analysis aimed to explore the association between formal social participation and cognitive function in middle-aged and older adults using data from longitudinal studies. A comprehensive search was conducted in Scopus, PubMed, and Web of Science for longitudinal studies that assessed the association between formal social participation and cognitive function in middle-aged and older adults published between January 2010 to 19 August 2022. Risk of bias was judged using the RoBANS tool. Meta-analysis using a random-effects model was computed with odds ratio (OR) and 95% confidence interval (CI) for cognitive decline probability. Sensitivity analyses were made to explore any changes to the pooled statistical heterogeneity and pooled effect size. Certainty of evidence was judged using the GRADE framework. We included 15 studies comprising 136,397 participants from 5 countries. Meta-analyses showed that formal social participation was associated with reduced cognitive decline (OR = 0.78, 95% CI 0.75–0.82, *p* < 0.001), with very low certainty of evidence. Formal social participation appears to enhance cognition in middle-aged and older adults, but further high-quality research is needed given the very low certainty of evidence.

## 1. Introduction

Cognitive decline has received increasing attention in recent years due to its highly individual and socio-economic impact. Scientific evidence links cognitive decline with a higher risk of progressing to mild cognitive impairment and dementia [1], elevated mortality risk among older adults [2,3], increased comorbidities [4], and lower quality of life due to daily activity limitations [1]. This decline is also associated with greater healthcare resource utilization [5] and places a significant burden on family members, particularly caregivers, who are more prone to depression, stress, and worse cognitive function [6,7]. Therefore, understanding the mechanisms capable of minimizing or postponing cognitive decline is key to improving older adults’ and families’ quality of life and alleviating healthcare system strain. It is widely acknowledged that cognitive decline is influenced by a dynamic interplay of genetic, biological, psychological, and social factors over the life course [8]. One such modifiable factor is social participation, recognized for its potential to mitigate cognitive decline [8,9,10,11,12,13].

While specific definitions may vary [14,15], social participation is usually classified into two types: informal, involving interactions with close social ties in an informal setting, and formal, characterized by interactions in established organizations [16,17]. Formal social participation is most commonly assessed through volunteering and involvement in clubs, groups, or associations.

Longitudinal studies focusing specifically on the association between formal social participation and cognition highlight that engaging in formal social activities benefits cognition for middle-aged and older adults, leading to higher cognitive scores and reduced risk of cognitive decline [9,18,19,20,21,22,23,24,25,26].

There is well-established evidence regarding the pathways through which formal social participation can positively influence cognition. Formal social participation may enhance cognitive function through a blend of cognitive [26] and physical engagement [19]. Additionally, it fosters the development of diverse and long-lasting social networks [27,28]. In turn, these networks may lead to increased social support [29], healthier behaviors [12,30], and a more socially integrated lifestyle [31], ultimately contributing to better cognition.

Despite solid longitudinal evidence of the benefits of formal social participation on cognition, some studies have failed to find such a connection [32,33], while others observed a significant relationship only for women [34].

To our knowledge, existing systematic reviews and meta-analyses on the association between social participation and cognition lack a specific focus on formal social participation, frequently combining informal and formal social participation and sometimes collapsing them into a single index [10,12,13]. Furthermore, one potential limitation of studies focusing on formal social activity and cognitive decline is the possibility of reverse causality, where individuals experiencing cognitive decline may already have reduced or ceased participation in formal social activities [13,21]. Therefore, upon the limitations of current systematic reviews, we aim to conduct a systematic review with meta-analysis of longitudinal studies to examine the association between formal social participation and cognitive function. This systematic review with meta-analysis will add to the existing body of literature by (1) focusing solely on formal social participation and (2) including only studies that excluded participants with cognitive impairment or dementia from baseline or assessed the impact of reverse causality in their findings.

## 2. Materials and Methods

This systematic review with meta-analysis followed the Preferred Reporting Items for Systematic Reviews and Meta-analysis (PRISMA) 2020 guidelines [35] (Appendix A). The protocol was developed a priori and registered in the International Prospective Register of Systematic Reviews (PROSPERO) with the code CRD42022342136.

### 2.1. Eligibility Criteria

We established the eligibility criteria based on the Participants, Interventions, Comparators, Outcomes, and Study Design (PICOS) framework. Participants: individuals aged 45 years and older without baseline cognitive impairment or dementia (as reported in the original study) unless reverse causality was assessed. Intervention: formal social participation is clearly defined and measured. Studies focusing solely on informal social participation or combining formal and informal social participation in a single index were excluded. Comparators: different levels of formal social participation or participation versus non-participation in formal social activities. Outcomes: global cognitive function as a dichotomous outcome (categorized as cognitive impairment or no cognitive impairment; cognitive decline or no cognitive decline) and global cognitive function as a continuous outcome (measured using cognitive function scores). We accepted valid measures reporting a global cognitive score across multiple domains, with no outcome measurement frequency restrictions but requiring a minimum one-year follow-up. Studies primarily focused on a single cognitive dimension (e.g., verbal fluency) or dementia as the main outcome were excluded. Study design: longitudinal prospective cohort studies with at least one year of follow-up, excluding qualitative or non-prospective designs. Given the unavailability of translation services, we limited our inclusion criteria to studies written in English, Portuguese, Spanish, Italian, or French.

### 2.2. Search Strategy

We systematically searched three electronic databases (PubMed, Scopus, and Web of Science) to identify longitudinal studies assessing the association between formal social participation and cognitive function in middle-aged and older adults. The search encompassed studies published between January 2010 and 19 August 2022 (Appendix A). Starting from 2010 allowed us to balance comprehensiveness and manageability, given the substantial number of articles in our field. This timeframe ensured we captured the most recent and up-to-date findings, as previous systematic reviews [10,12,13] have already covered earlier studies. We also screened the reference list of relevant reviews and included studies to identify potential articles not initially found through the database searches.

### 2.3. Study Selection and Data Extraction

The resulting studies from database searches were organized into a Microsoft Excel sheet and the duplicate results were removed by manual screening. All records were revised by two authors according to the eligibility criteria. Disagreements were resolved by consensus.

Once the selection of studies was finalized, the data were extracted by two researchers using a standardized form on Microsoft Excel and checked by the senior author. Data extraction focused on the following information about the studies, populations, outcomes measures, and results: (1) year of publication and country; (2) data source; (3) follow-up in years; (4) sample size; (5) baseline sociodemographic characteristics (mean age, age range, and sex distribution); (6) description of the cognitive function measures; (7) description of the formal social participation measure; (8) adjusted confounders; and (9) main findings reported in the original articles.

### 2.4. Risk of Bias

Two researchers judged the risk of bias using the Risk of Bias Assessment tool for Non-randomized Studies (RoBANS) [36], which was cross-checked by the remaining authors. The RoBANS judges the risk of bias of non-randomized studies based on six bias domains: participant selection, confounding variables, exposure measurement, blinding of outcome assessment, incomplete outcome data, and selective outcome reporting. Two authors judged each domain as low risk, high risk, or unclear risk of bias. There is no specific tool for bias assessment in studying the impact of formal social participation on cognition in middle-aged and older adults and, therefore, we adapted the RoBANS tool to suit our research topic (Appendix A).

### 2.5. Data Synthesis

All analyses were performed using Rstudio software (version 4.0.5) and the “meta” package. Following recommended guidelines [13,37], we dichotomized categorical presentations of formal social participation. This involved comparing the lowest category, representing no formal participation or low levels, against the combined remaining higher categories. To ensure consistency in integrating study results into the meta-analysis, we applied the established method of 1/OR (odds ratio) [10] to studies with dichotomous cognitive function outcomes, harmonizing the results in the same direction.

When studies separately reported outcomes for women and men, we included both unless findings for the mixed population were also provided. When studies presented results for the entire sample as well as results divided into age cohorts, we prioritized the inclusion of results concerning the whole sample.

The meta-analysis was performed using a random-effects model to mitigate the effect of different sources of heterogeneity (differences in outcome metrics and how these were collected) in the true effects [37]. The DerSimonian and Laird estimator [38] was used to estimate between-study variance components. The impact of statistical heterogeneity was assessed using I^2^ statistics [39]: values below 50% indicated low heterogeneity, values between 50% and 75% were classified as having moderate heterogeneity, and those exceeding 75% were considered indicative of high heterogeneity. In our analysis, we reported results as odds ratios (ORs) with 95% confidence intervals (CIs) for dichotomous outcomes.

We conducted a series of post hoc sensitivity analyses to explore any changes to the pooled statistical heterogeneity and pooled effect size. The following sensitivity analyses were performed by sequentially removing studies (1) with the lowest (outlier) mean age; (2) with a large sample size (n > 6809); (3) in which formal social participation was assessed for a specific focused activity (e.g., voluntary work) rather than a broader set of formal social activities; and (4) with partial overlapping of samples (the study with the largest sample size was retained). The sensitivity analyses were conducted with a minimum criterion of retaining at least two studies.

### 2.6. Certainty of Evidence

The certainty of evidence was judged by two authors using the Grading of Recommendations Assessment, Development, and Evaluation (GRADE) approach [40]. The certainty of evidence was graded as high, moderate, low, or very low. Downgrading of certainty occurred when there were concerns regarding the risk of bias, inconsistency, imprecision, and indirectness. The risk of publication bias was not assessed in our meta-analysis due to the limited number of studies available [41].

## 3. Results

### 3.1. Search Results and Study Characteristics

The electronic database search identified 32,798 published articles, of which 11,889 were duplicated and dropped. No relevant articles were found in the reference lists of the included studies. The remaining 20,909 titles and abstracts were screened, resulting in 143 being selected for full-text analysis. A total of 15 observational studies met the eligibility criteria and were incorporated into this systematic review [18,20,21,22,23,24,25,32,33,34,42,43,44,45,46] (Figure 1). Among these, six studies were further included in the quantitative synthesis [24,25,33,34,43,44].

Table 1 presents the characteristics of all 15 articles included in the analysis. These studies were conducted between 2010 and 2022, with 11 studies conducted in Asian countries, 3 in the United States of America, and 1 in Brazil. The total sample consisted of 136,397 middle-aged and older adults, who were followed for an average period of 10 ± 4.2 years. The participants had a mean age of 68 ± 5.8 years, and in 9 out of 15 studies, women accounted for more than half of the sample. Moreover, in 14 studies, precautions were taken to reduce the potential influence of reverse causality by excluding individuals with abnormal cognitive performance and dementia at baseline. Although one study [21] did not implement this exclusion, an assessment of the impact of reverse causality on the final result was conducted and determined as not significant.

### 3.2. Risk of Bias

All included studies were judged as high risk in at least one bias domain (Figure 2). Low selection bias stemming from participant selection was observed in all studies, while 40% of the studies demonstrated a high risk of selection bias due to uncontrolled confounding variables. More than half of the studies (53%) exhibited a high risk of performance bias due to insufficiently precise exposure measurement. Detection bias was considered high in nearly all studies (93%). Regarding attrition bias, most studies had a high (33%) and an unclear risk (27%). Lastly, all studies were judged to have a low risk of selective reporting bias.

### 3.3. Outcome Characteristics

All studies assessed the participants’ cognition during the follow-up period using tools that measure global cognitive function, such as the Mini-Mental State Examination (MMSE) [18,20,23,24,25,32,42,44,45], the Short Portable Mental Status Questionnaire (SPMSQ) [33], the Cognitive Performance Scale (CPS) [34], the Activities of Daily Living Independence Assessment Criteria for Elderly Individuals [22], and composite scores from various cognitive questions adapted from the Telephone Interview for Cognitive Status [21,43,46]. Cognitive function was dichotomous in eight studies [18,22,24,25,33,34,43,44] and continuous in seven studies [20,21,23,32,42,45,46].

### 3.4. Formal Social Participation Characteristics

The vast majority of studies considered several formal social activities, such as volunteering, participating in sports clubs, political organizations, and religious organizations, among other activities [18,23,24,25,32,34,42,44,45]. Four studies [20,21,43,46] focused solely on formal volunteering, while Yen et al. [33] focused only on joining an organized group activity and Tsuji et al. focused on community-level sports group participation [22].

More than half of the studies (53%) assessed the participation in formal social activities only in baseline [20,23,24,32,33,34,44,45] and almost three-fourths of the studies (73%) examined different levels of formal social participation (e.g., participation scores, change in the likelihood of participation, the total number of social activities −0, 1, ≥2) [20,21,23,24,25,34,42,43,44,45,46].

### 3.5. Confounding Variables

The included studies addressed various confounding factors to minimize potential biases and enhance the validity of their findings. Common confounding variables included age, sex, education, marital status, income, depressive symptoms, and functional limitations.

### 3.6. Quantitative Synthesis on the Dichotomous Measurement of Cognitive Function

The meta-analysis showed that engaging in formal social activities significantly decreases the likelihood of a decline in cognitive abilities (OR = 0.78, 95% CI: 0.75 to 0.82, *p* < 0.001; very low certainty of evidence; Figure 3), even when considering the prediction interval (95% PI: 0.73 to 0.84). The analysis revealed low non-significant heterogeneity (I^2^ = 0%, *p* = 0.45). Sensitivity analyses (Appendix A) showed that overall heterogeneity remained low, and the pooled effect consistently maintained statistical significance. Nevertheless, upon considering the prediction interval, the pooled effect lost statistical significance when excluding combined studies with larger sample sizes [25,43].

## 4. Discussion

### 4.1. Findings in the Context of Existing Knowledge

To our knowledge, this is the first meta-analysis investigating the longitudinal association between formal social participation and cognitive function in middle-aged and older adults. To address potential issues with reverse causality, we followed Piolatto’s [13] suggestion by including studies that excluded participants with cognitive impairment or dementia at baseline or explored the influence of reverse causality on their results.

The meta-analysis revealed an overall significant association between participating in formal social activities and improved cognition. Despite the statistically significant effects, we advise some caution when interpreting our findings, since the meta-analysis had a very low certainty of evidence, which is attributed to the risk of bias and indirectness in the included studies. The meta-analysis was also performed with a small number of studies; however, the pooled sample size was very large, mitigating the concerns about imprecision. Nevertheless, the meta-analysis showed no heterogeneity, and the results remained consistent and with low heterogeneity even after sensitivity analysis. Therefore, despite the very low certainty of evidence, these results strengthen our confidence in the observed association between formal social participation and lower cognitive decline or impairment.

Previous systematic reviews, with or without meta-analysis, consistently point to the cognitive benefits of social participation [10,12,13]. However, it is essential to note that these systematic reviews focused on the combined effects of formal and informal social participation. In contrast, our systematic review specifically isolated the impact of formal social participation, allowing for a targeted investigation into its association with cognition. By disentangling the effects of formal social activities, we offer unique insights that contribute to a more nuanced understanding of the specific benefits of formal social participation.

Formal social participation may offer benefits for cognitive function due to a combination of factors, including mental [26] and physical stimuli [19] provided by these activities, as well as the social component that fosters the formation of long-lasting and diverse social networks [27,28]. Such networks may lead to increased social support [29], the adoption of healthy behaviors [12,30], or a more socially integrated lifestyle [31].

It is also important to highlight that 53% of the studies included in our systematic review solely assessed formal social participation at baseline. This observation underscores the importance of exploring the dynamic changes in social activity patterns over time to attain a more comprehensive understanding of its cumulative impact on cognition. Therefore, future research should prioritize exploring repeated measurements of formal social participation and its association with cognition. Additionally, it is noteworthy that the majority of the analyzed studies overlooked age group differences in the association between formal social participation and cognition. Recognizing potential age-related variations in this association is crucial, as it can uncover distinct cognitive trajectories across different age cohorts and inform tailored interventions and targeted policy recommendations. As we move forward, it is important that future studies explicitly consider age group differences when investigating the relationship between formal social participation and cognition.

Notwithstanding the limitations in the literature, the findings of our systematic review are encouraging. There was a strong link between social participation and a lower likelihood of decline in cognitive function with data from 33,489 participants. These findings should encourage researchers and funders to pursue further research to identify important key components of social participation that can further enhance cognitive function as well as prompt policy makers to promote formal social participation education and intervention strategies amongst all relevant stakeholders (institutionalized and non-institutionalized older adults, their families and caregivers, and all other professionals that intervene with older people).

### 4.2. Limitations

Several limitations should be acknowledged. Although we found associations between formal social participation and cognition, the very low certainty of evidence underscores the need for additional longitudinal studies to investigate this association further. Moreover, despite the fact that our approach improved upon previous meta-analyses by focusing specifically on formal social participation, which reduced some diversity within the exposure variable, variability still exists due to the range of activities and assessment methods, possibly leading to potential relevant research oversights. Additionally, different methodologies assessing formal social participation and cognitive function may have contributed to indirectness, limiting generalizability, which we addressed by employing a random-effects model. There was also an overrepresentation of studies from Asian countries, making it interesting to assess the effect of the geographic area in the subgroup analysis due to cultural differences. However, the limited number of studies in the other regions prevented this. Furthermore, although there were three studies with partially overlapping samples, the sensitivity analysis revealed no statistically significant changes in the effect size. The main meta-analysis retained all three studies because, despite using data from the same database, each study displayed distinct mean sample characteristics, encompassing variations in age, sex, follow-up duration, and covariates.

Another limitation was the inability to conduct quantitative synthesis on the continuous measurement of cognitive function. This challenge stems from the considerable heterogeneity in assessment instruments used across studies where cognition is treated as a continuous outcome, necessitating the reporting of results as standardized mean differences. A critical obstacle faced in this process was the inadequate reporting of necessary details to perform this meta-analysis. This limitation highlights the importance of future research to prioritize a more uniform and detailed reporting approach.

## 5. Conclusions

We found a statistically significant association between formal social participation and the cognitive function of middle-aged and older adults, which policymakers should not overlook. However, these results should be interpreted with a grain of salt as these are based on very low certainty of evidence. This study also highlights the need for further longitudinal research on the association between formal social participation and cognitive function, particularly in countries where this has not been investigated. Future studies should also explore the modifiable nature of formal social participation over time and consider age group differences when investigating the relationship between formal social participation and cognition. 

## Figures and Tables

**Figure 1 behavsci-14-00262-f001:**
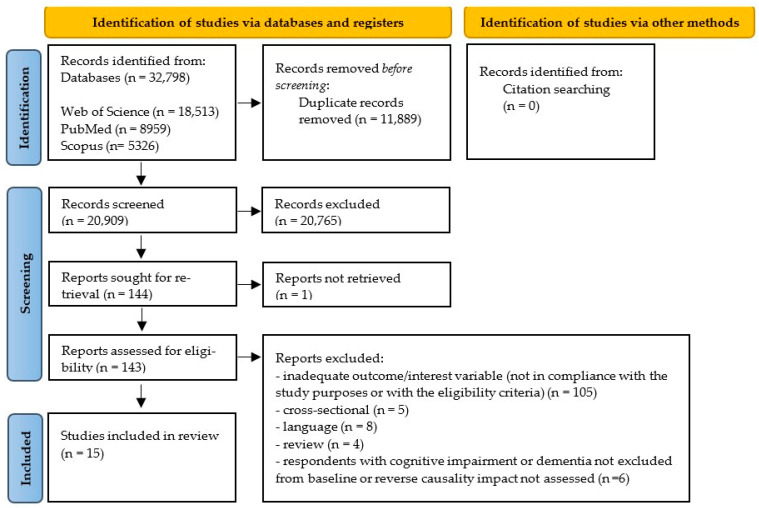
PRISMA flow diagram of the study selection process.

**Figure 2 behavsci-14-00262-f002:**
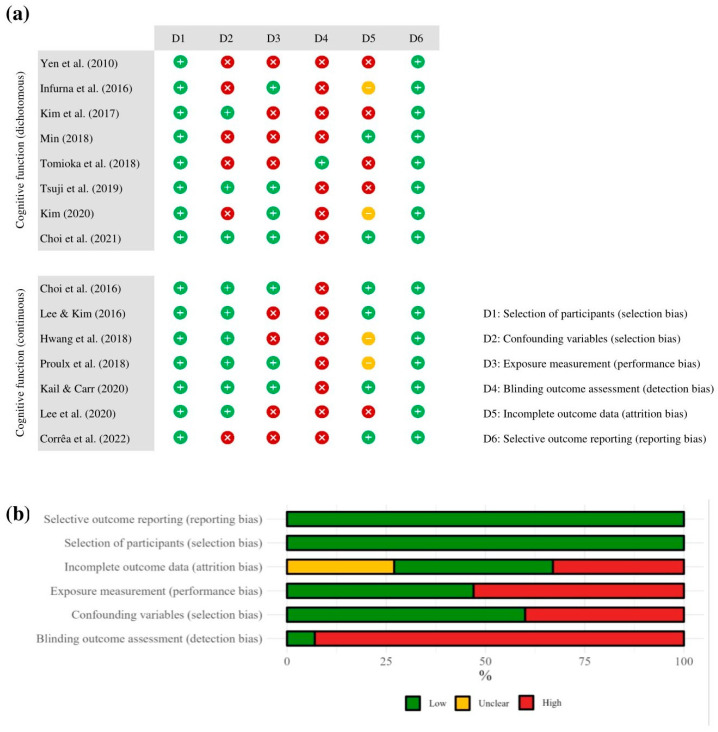
Risk of bias judgment: (**a**) study level; (**b**) overall summary of all included studies; [18,20,21,22,23,24,25,32,33,34,42,43,44,45,46].

**Figure 3 behavsci-14-00262-f003:**
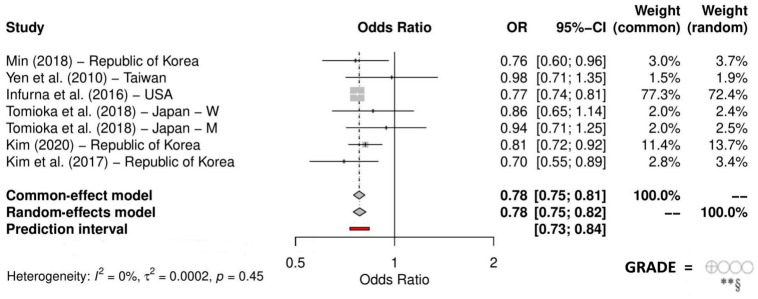
Quantitative synthesis on the dichotomous measurement of cognitive function. Notes: ** = Downgraded two levels due to very serious study limitations (high risk of bias in at least three domains); § = Downgraded one level due to serious indirectness (different methods for assessing formal social participation and cognitive function); [24,25,33,34,43,44].

**Table 1 behavsci-14-00262-t001:** Main characteristics of the included studies.

Author (Year), Country	Data Source	Follow-Up Duration	Sample Size	Baseline Participant Characteristics		Cognitive Function Assessment (Outcome of Interest)	Formal Social Participation Assessment	Adjusted Confounders	Main Findings Reported in the Original Article
				Age (Mean ± SD; Range)	% Women				
Yen et al. (2010) [33], Taiwan	SHLSET	10	1626	69.8 ± 4.9; 60+	59.2	Dichotomous (no cognitive impairment/cognitive impairment); Measured with SPMSQ.	Dichotomous (no/yes); Assessment: joining an organized group activity; Assessed at baseline.	Age, sex, marital status, education, ethnicity, smoking, alcohol drinking, depression, hypertension, diabetes, cardiovascular disease, stroke, ADL disability, IADL disability, functional limitation, and self-perceived health.	Participating in organized group activities did not show any significant association with a decreased likelihood of cognitive impairment.
Infurna et al. (2016) [43], USA	HRS	14	13,262	71.3 ± 8.3; 60–106	58	Dichotomous (no cognitive impairment/cognitive impairment); Measure adapted from TICS.	Continuous variable; Assessment: change in the likelihood of volunteering; Assessed over time.	Age, sex, ethnicity, marital status, education, employment status, physical exercise, smoking, functional limitations, self-rated health, cardiovascular illnesses, and depressive symptoms.	Volunteering regularly over time was significantly associated with a decreased likelihood of cognitive impairment.
Kim et al. (2017) [44], Republic of Korea	KLoSA	6	2495	71.2; 65–79	53.39	Dichotomous (no cognitive decline/cognitive decline); Measured with K-MMSE.	Variable levels: 1, 2, 3+ activities; Assessment: religious groups; senior café; sports and leisure clubs; alumni groups; volunteering; political and non-governmental organizations; Assessed at baseline.	Age, sex, education, employment status, depression, IADL, and weight loss.	Greater participation in formal social activities was significantly associated with a decreased likelihood of cognitive decline.
Min (2018) [24], Republic of Korea	KLoSA	6	2445	67.5 ± 5.6; 60–91	46	Dichotomous (no cognitive decline/cognitive decline); Measured with K-MMSE.	Continuous variable (0 to 6); Assessment: church/religious groups, social clubs, sports clubs, alumni societies, volunteer groups, and political organizations; Assessed at baseline.	Age, sex, marital status, and education.	Increased participation in formal social activities was significantly associated with a decreased likelihood of cognitive decline.
Tomioka et al. (2018) [34], Japan	------	3	6093	72.8; 65–96	54.6	Dichotomous (no cognitive decline/cognitive decline); Measured with CPS.	Variable levels: 0, 1, 2, 3+ activities; Assessment: neighborhood associations, hobby groups, local event groups, senior citizen clubs, and volunteer groups; Assessed at baseline.	Age, family structure, BMI, pensions, number of medications used, self-reported medical conditions, alcohol consumption, smoking, depression, self-rated health, and IADLs.	In women, participating in three or more formal social activities was significantly associated with a decreased likelihood of cognitive decline, while no other associations were found.
Tsuji et al. (2019) [22], Japan	JAGES	6	40,308	65+	51.31	Dichotomous (no cognitive impairment/cognitive impairment); Measured with The Activities of Daily Living Independence Assessment Criteria for Elderly Individuals.	Dichotomous (frequency: 1+ day per month; no participation); Assessment: sports group participation. Assessed over time.	Age, sex, population density, annual sunlight hours, stroke, hypertension, diabetes, hearing loss, obesity, social isolation, drinking status, smoking status, education, income, depression, and walking time.	Higher prevalence of sports group participation showed a statistically significant relationship with lower risk of cognitive impairment.
Kim (2020) [25], Republic of Korea	KLoSA	10	7568	56.01; 45+	48.16	Dichotomous (no cognitive impairment/cognitive impairment); Measured with K-MMSE.	Variable levels: participating never or almost never, once or twice a month, almost every week+; Assessment: religious, friendship, leisure/sports, alumni, volunteer, and political associations; Assessed over time.	Age, sex, education, income, employment, marital status, location, physical activity, contact with friends, smoking, hypertension, diabetes, cerebrovascular and heart diseases, obesity, hearing loss.	Participating in formal social activities was significantly associated with a decreased likelihood of cognitive impairment.
Choi (2021) [18], Republic of Korea	KLoSA	12	7568	45+	50.92	Dichotomous (no cognitive decline/cognitive decline); Measured with K-MMSE.	Dichotomous (socially active: 1+ activity; socially inactive: none); Assessment: religious, social, sports/cultural/leisure groups, college programs, alumni groups, grand family associations, volunteering, political party, civil organizations, and interest groups. Assessed over time.	Age, sex, education, income, smoking, alcohol consumption, comorbidity, and depression.	Not participating in formal social activities showed a greater risk of overall cognitive decline.
Choi et al. (2016) [42], Republic of Korea	KLoSA	6	6076	58.5 ± 9.5; 45–93	50.8	Continuous (higher scores = better cognition); Measured with K-MMSE.	Variable levels: consistent non-participation, participation to no participation, no participation to participation, consistent participation; Assessment: religious, friendship, political organizations, leisure/ culture/sports clubs, family/school reunions, volunteer work; Assessed over time.	Age, sex, marital status, education, income, employment status, number of chronic diseases, regular exercise, and area of living.	The non-participation to participation and consistent participation groups exhibited significantly higher cognitive function scores than those with inconsistent participation in formal social activities.
Lee & Kim (2016) [45], Republic of Korea	KLoSA	4	1568	71.06 ± 0.12; 65+	45.7	Continuous (higher scores = better cognition); Measured with K-MMSE.	Variable levels: 0, 1, 2+ activities; Assessment: church or other religious groups; senior citizen clubs or senior centers; alumni societies or family councils; Assessed at baseline.	Age, sex, marital status, education, household income, living arrangement, residential area, comorbidities, ADL, IADL, quality of life, and depressive symptoms.	No significant association was found between participating in formal social activities and a decline in cognitive function scores.
Hwang et al. (2018) [32], Republic of Korea	KLoSA	8	6706	58.1 ± 0.12; 45+	50.1	Continuous (higher scores = better cognition); Measured with K-MMSE	Dichotomous variable (no/yes for each activity); Assessment: religious groups; social gatherings; alumni/clan gatherings; and volunteer work; Assessed at baseline.	Age, sex, education, income, employment status, marital status, region of residence, physical activity, smoking, alcohol use, limited activities of daily living, depression, and comorbidity.	No significant association was found between participating in formal social activities (all types) and cognitive function scores.
Proulx et al. (2018) [46], USA	HRS	16	11,100	64.68 (0.12); 50+	51.97	Continuous (higher scores = better cognition); Measure adapted from TICS.	Variable levels (hours volunteering in the past 12 months): 0 h, 1–99 h, 100–199 h, 200+ hours; Assessment: formal volunteering; Assessed over time.	Age, sex, race, ethnicity, marital status, household income, household wealth, IADL, self-rated health, depressive symptoms, and time.	Participating in formal volunteering is significantly associated with higher cognitive function scores.
Kail & Carr (2020) [21], USA	HRS	16	27,485	66.66 ± 9.91; 50–105	59.8	Continuous (higher scores = better cognition); Measure adapted from TICS.	Variable levels: no volunteering; low level; moderate level; and high level; Assessment: formal volunteering; Assessed over time.	Age, sex, ethnicity, education, marital status, self-rated health, disability, symptoms of depression, income, household wealth, and hours working per week.	A small portion of cognitive benefits from formal volunteering stems from volunteers’ higher cognitive levels. However, older volunteers maintain higher cognitive function scores even after accounting for this.
Lee et al. (2020) [23], Republic of Korea	KLoSA	10	1806	65+	-----	Continuous (higher scores = better cognition); Measured with K-MMSE.	Variable levels: 0, 1, 2+ activities; Assessment: religious groups; friendship clubs; leisure or sports clubs; and political organizations; Assessed at baseline.	Age, employment, region of residence, marital status, living arrangement, education, income, chronic disease, depression, and disability status.	Participating in formal social activities is significantly associated with higher cognitive function scores in both men and women.
Corrêa et al. (2022) [20], Brazil	------	2	291	69.64 ± 6.38; 65+	89.4	Continuous (higher scores = better cognition); Measured with MMSE.	Continuous variable (number of days volunteered last month); Assessment: formal volunteering; Assessed at baseline.	Age, sex, marital status, education, family income, race, smoking, physical health, and mental health.	No significant association was found between volunteering and cognitive function scores.

Notes: USA = United States of America; SHLSET = Survey of Health and Living Status of the Elderly in Taiwan; HRS = Health and Retirement Study; KLoSA = Korean Longitudinal Study of Aging; JAGES = Japan Gerontological Evaluation Study; SPMSQ = Short Portable Mental Status Questionnaire; TICS = Telephone Interview for Cognitive Status; K-MMSE = Korean Mini-Mental State Examination; CPS = Cognitive Performance Scale; ADL = Activities of Daily Living; IADL = Instrumental Activities of Daily Living; BMI = Body Mass Index.

## Data Availability

Data are available in a public, open access repository: https://osf.io/zh5dt/, made public on 16 February 2024.

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
