# Peer review of "Is Formal Social Participation Associated with Cognitive Function in Middle-Aged and Older Adults? A Systematic Review with Meta-Analysis of Longitudinal Studies"

_behavsci, 2024, doi:10.3390/bs14040262_

Round 1
Reviewer 1 Report
Comments and Suggestions for Authors
The study conducts a systematic review and meta-analysis about the association between the formal social participation and the cognitive function. This meta-analysis is adapted to the PRISMA standards and its content is interesting although there is a part of the methodology that should be explained in more detail.
In this article the different sections of a systematic review and meta-analysis are well explained in general. The problem is related to the integration of quantitative data. In the case of qualitative data there is no problem because the odds ratio is well determined when the data are binary, in fact, the authors do not find heterogeneity in the integration of odds ratios. However, when the data are quantitative, the authors choose as effect size the difference in means estimated from the beta coefficient of a multilevel model or mixed model. The first limitation is due to the fact that the articles do not use the same instrument to assess cognitive function. Therefore, mean differences cannot be used as an effect size when the scales are different. In this case, the standardized mean difference proposed by Cohen should be used. Although the great limitation that the authors have with this type of articles is that the mean difference has been estimated from a linear mixed model and the articles do not show the standardized betas that would be the effect sizes that would have to be integrated. Another problem that arises due to using the betas of a model is that these betas are adjusted for different covariates depending on the articles, which produces an important source of variability that should not be accounted for by the random effects model.
Another important limitation of using the betas from mixed models is that some articles do not provide standard errors and they are estimated according to the proposal described in "Higgins, J. P., & Green, S. (Eds.). (2008). Cochrane handbook for systematic reviews of interventions." Higgins, J. P., & Green, S. (Eds.). (2008). Cochrane handbook for systematic reviews of interventions”. This proposal is valid when using mean differences from a classic linear model where the estimation of Beta (difference of means) is carried out by the least squares method. However, in this meta-analysis, the betas come from mixed or multilevel models where the restricted maximum likelihood method is used to estimate them, therefore the calculation of the standard error proposed in the “Data synthesis” section is not valid for the models used. Collecting information from more complex models is difficult because the studies generally do not provide enough information to get a reliable effect size that can be integrated with others. They provide so much variability that the integrated effect size could be anywhere from 0 to a value so high that it would have no experimental interpretation.
In short, the validity of the integration of effect sizes with quantitative data is very poor and it would therefore be advisable to eliminate it from the study or try to obtain the means and standard deviations of the articles that provide them. The authors have to take into account that the data are longitudinal and therefore possibly dependent. In the literature there are several articles describing the calculation of the standardized mean difference in this context.
Other minor comments are:
1.- The classification of the impact of heterogeneity as low, moderate and high based on the I2 coefficient is discouraged by Borenstein in his article “Borenstein, M. (2023). Avoiding common mistakes in meta‐analysis: Understanding the distinct roles of Q, I‐squared, tau‐squared, and the prediction interval in reporting heterogeneity. Research Synthesis Methods.”. Borenstein indicates that to assess how effect sizes vary, it is best to calculate the prediction interval.
2.- The authors make no mention of the problem of dependence on effect sizes when there are effect sizes that come from the same study and there are studies that have the same author and possibly share sample size. Hedges & Olkin (2014) warns of this problem because the homogeneity test is based on the assumption that the data are independent. It would be interesting to at least mention it in the limitations.
(Hedges, L. V., & Olkin, I. (2014). Statistical methods for meta-analysis. Academic press.)
Reviewer 2 Report
Comments and Suggestions for Authors
Thank you for the opportunity to review the study. Overall, it’s a very interesting and I read with enthusiasm. From another perspective, I think it’s also well-written and methodologically rigorous.
I’d like to propose just some very quick-to-solve changes.
- Specifically, I’d highly suggest including p- alongside meta-analysis results, starting from the Abstract to the ‘results’ section;
- The introduction maintains consistency but relies heavily on outdated refs. (> 70 %). To align with the current state of the field, and best practices, it’s essential updating most, if not all, refs;
- Additionally, please provide HQ-illustration for improved readability, and consider tidying Tables when necessary;
- Also, it’s important the authors check if residuals caused overfitting, i.e when pooled / calculating weight or effects from study;
- Finally, apart from the ‘obvious’ encouraging placing data on OSF, I think the authors could place any calculation or any sort of different thing they did in this study. Why? With this, authors can create some kind of “side chain” disseminating study at the same time share the how-to.
Wishing you success with the study.
Comments on the Quality of English LanguageJust double-check sentences and refs. list
Round 2
Reviewer 1 Report
Comments and Suggestions for Authors
I consider that the authors have taken into account the proposed changes and I thank them for their explanations regarding these changes. In my opinion this systematic review meets the quality standards to be published.
Author Response
Dear Reviewer,
We appreciate your positive feedback on the revisions made in response to the first round of comments. It's encouraging to note that you find that our systematic review meets the quality standards to be published. Thank you for the thoroughness of your review and the time you dedicated to providing valuable insights for our manuscript.